

# Molecular characterization and expression profiling of *transformer 2* and *fruitless*-like homologs in the black tiger shrimp, *Penaeus monodon*

Prawporn Thaijongrak[1,2,3], Charoonroj Chotwiwatthanakun[1,4], Phaivit Laphyai[1,2], Anuphap Prachumwat[1,5], Thanapong Kruangkum[1,2], Prasert Sobhon[2] and Rapeepun Vanichviriyakit[1,2]

[1] Center of Excellence for Shrimp Molecular Biology and Biotechnology, Faculty of Science, Mahidol University, Bangkok, Thailand
[2] Department of Anatomy, Faculty of Science, Mahidol University, Bangkok, Thailand
[3] Department of Clinical Sciences and Public Health, Faculty of Veterinary Science, Mahidol University, Nakhon Pathom, Thailand
[4] Nakhonsawan Campus, Mahidol University, Nakhonsawan, Thailand
[5] National Center for Genetic Engineering and Biotechnology, National Science and Technology Development Agency, Pathum Thani, Thailand

Corresponding author
Rapeepun Vanichviriyakit,
rapeepun.van@mahidol.edu

## ABSTRACT

*Transformer 2* (*tra 2*) and *fruitless* (*fru*) genes have been proven to play a key role in sex determination pathways in many Arthropods, including insects and crustaceans. In this study, a paralog of *P. monodon tra 2 (Pmtra 2)*, *P. monodon ovarian associated transformer 2* (*PmOvtra 2*) and 2 isoforms of *P. monodon fruitless*-like gene (*Pmfru-1* and *Pmfru-2*) were identified and characterized. The full cDNA sequence of *PmOvtra 2* consisted of 1,774 bp with the longest open reading frame (ORF) of 744 bp encoding for 247 amino acids. The PmOvtra 2 exhibited a predicted RNA-recognition motif (RRM) domain and two arginine-serine (RS) regions, suggesting its function in RNA splicing. The full cDNA sequence of *Pmfru-1* consisted of 1,306 bp with 1,182 bp ORF encoding for 393 amino acids, whereas the full cDNA sequence of *Pmfru-2* consisted of 1,858 bp with 1,437 bp ORF encoding 478 amino acids. The deduced amino acid sequences of Pmfru-1 and Pmfru-2 exhibited highly conserved domains of Fru proteins, including Broad-complex, Tramtrack and Bric-a-brac (BTB), and zinc finger (ZF) domains. In addition, *Pmfru-1* and *Pmfru-2* were suggestively originated from the same single genomic locus by genomic sequence analysis. Specifically, *Pmfru* pre-mRNA was alternatively spliced for *Pmfru-1* and *Pmfru-2* to include mutually exclusive exon 7 and exon 6, respectively. Temporal and spatial expression of *PmOvtra 2*, *Pmfru-1,* and *Pmfru-2* were also investigated by qPCR. The results showed that all were expressed in early developmental stages with undifferentiated gonads starting from nauplius until postlarvae. The expression of *PmOvtra 2* started at nauplius stage and gradually increased from mysis to postlarvae (PL) 1. However, the expression of *Pmfru-1* was low at the nauplii stage and slightly increased from protozoea to PL5, whereas the expression of *Pmfru-2* maintained a low level from nauplius to mysis and then gradually increased at the PL stages. Expressions of *PmOvtra 2, Pmfru-1,* and *Pmfru-2* were detected in various tissues including nervous tissue, gill, heart, hepatopancreas, gut, and gonads. Interestingly, the sexually dimorphic expression of *PmOvtra 2, Pmfru-1,*

and *Pmfru-2* was demonstrated in fully developed gonads in which the ovary showed significantly higher expressions than the testis. The great difference in the expression pattern of *PmOvtra 2, Pmfru-1,* and *Pmfru-2* in the ovary and testis suggested their roles in the female sex determination in *P. monodon*.

## INTRODUCTION

Shrimp farming is a major aquaculture sector in the world that generates high economic value. *Penaeus monodon* or black tiger shrimp is one of the main aquaculture species widely distributed in the Indo-Pacific region (*Flegel, 2012*; *Mandal et al., 2012*). It belongs to the family Penaeidae, which is known for its large size and fast growth rate. In crustacea, many species generally show sexual dimorphism in growth, especially *Macrobrachium spp.* and *Penaeus spp.* (*Bajaniya et al., 2014*; *Nogueira et al., 2019*). *Penaeus monodon* also exhibits differential growth rate in which the female shows superior growth to male. Therefore, culturing an all-female monosex population is highly attractive to increase the production of shrimp (*Li et al., 2012*; *Mareddy et al., 2011*). Phenotypic differences between males and females are thought to principally result from sex determination and sex differentiation gene expressions (*Liu et al., 2015*). Therefore, the molecular and genetic pathways of sex determination and sex differentiation in shrimp have been receiving great attention in recent years.

Mechanisms of sex determination in animals have been classified as either environmental sex determination (ESD) or genotypic sex determination (GSD) (*Nöthiger & Steinmann-Zwicky, 1985*; *Sánchez, 2004*). The mechanisms of sex determination in shrimp have been reported to be GSD with ZW sex determination system, in which female are heterogametic (ZW) and male are homogametic (ZZ) (*Li et al., 2003*; *Staelens et al., 2008*; *Zhang et al., 2007*). However, little is known about the pathway of sex determination in penaeid shrimp.

The molecular mechanisms of sex determination in insects have been well characterized. In *Drosophila*, the primary signal of sex determination is sex lethal (Sxl) protein (*Penalva & Sánchez, 2003*). The Sxl protein acts as the master switch that controls female-specific splicing of pre-mRNA of *transformer* (*tra*) (*Inoue et al., 1990*; *Penalva & Sánchez, 2003*). Dimerization of Tra protein and Transformer-2 (Tra-2) protein regulate female sex determination in *Drosophila* (*Nagoshi et al., 1988*) but functional Tra is not expressed in males (*Boggs et al., 1987*). As a splicing factor, Tra-2 consists of RNA-recognition motif (RRM) domain and two arginine-serine (RS) regions. The Tra and Tra-2 heterodimer functions in sex-specific alternative splicing of downstream RNA including *doublesex* (*dsx*) and *fruitless* (*fru*) (*Heinrichs, Ryner & Baker, 1998*; *Inoue et al., 1992*; *Penalva & Sánchez, 2003*; *Pomiankowski, Nöthiger & Wilkins, 2004*). *The* Dsx and Fru proteins act as transcription factors which control development of sex-specific differences in many insects (*Pomiankowski, Nöthiger & Wilkins, 2004*). The *dsx* is also involved in somatic

sex differentiation and some aspect of sex behavior (*Burtis & Baker, 1989*; *Christiansen et al., 2002*; *Pomiankowski, Nöthiger & Wilkins, 2004*). The *fru* plays a critical role in male courtship behavior and sex-nonspecific developmental functions (*Dauwalder, 2011*; *Demir & Dickson, 2005*; *Li et al., 2017*). The *fru* is a multifunctional gene having a complex coordination of four promoters and alternative splicing which can encode several isoforms of *fru* (*Dalton et al., 2013*). All fru isoforms exhibit Broad-complex, Tramtrack and Bric-a-brac (BTB) domain and zinc finger (ZF) domain. The BTB domain serves as a protein-protein interaction module, whereas the ZF domain serves as DNA-binding motif (*Salvemini et al., 2013*; *Zollman et al., 1994*).

Several insect homologs of sex determining genes have been identified and characterized in many decapod crustaceans such as *Penaeus vannamei* (*sxl*) (*López-Cuadros et al., 2018*), *Macrobrachium nipponense* (*sxl 1* and *sxl 2*) (*Zhang et al., 2013a*; *Zhang et al., 2013b*), *P. monodon (tra 2)* (*Leelatanawit et al., 2009*), *Penaeus chinensis* (*tra 2*) (*Li et al., 2012*), *Scylla paramamosain* (*tra 2*) (*Wang et al., 2020*), *Palaemon serratus* (*tra 2*) (*González-Castellano et al., 2019*), *Sagmariasus verreauxi* (*tra 2*) (*Chandler et al., 2015*), *Eriocheir sinensis* (*tra 2*) (*Luo et al., 2017*), *Daphnia magna* (*dsx*) (*Kato et al., 2011*), and *E. sinensis* (*fru1* and *fru2*) (*Li et al., 2017*). Most sex determining genes identified in crustaceans display high sequence similarity with their homologs in insects. In addition, homolog of *Caenorhabditis elegans* sex determining gene *feminization-1* (*fem-1*) has been reported in *P. vannamei* and *P. monodon* (*Galindo-Torres et al., 2019*; *Robinson et al., 2014*). However, the functions of sex determining genes in many crustaceans have not been investigated.

In this study, we identified and characterized sex determining homologs in *P. monodon* including *ovarian associated transformer 2 (PmOvtra 2)*, and 2 isoforms of *fruitless*-like gene (*Pmfru-1 and Pmfru-2*). Moreover, their temporal and spatial expression profiles were investigated. All of them showed sexually dimorphic expression patterns in gonadal tissues. Dominant expressions of *PmOvtra 2*, *Pmfru-1*, and *Pmfru-2* were shown in ovarian tissue, suggesting their role in female-sex determination in this species.

## MATERIALS AND METHODS

### Nucleotide sequences and deduced amino acid sequences analyses

Nucleotide sequences of *P. monodon* candidate sex determining genes *PmOvtra 2*, *Pmfru-1* and *Pmfru-2* were obtained from our transcriptome data prepared from *P. monodon* central nervous tissue (unpublished data). The nucleotide and deduced amino acid sequences of the putative sex determining genes were analyzed with Expasy Translate tool (*Gasteiger et al., 2003*) (https://web.expasy.org/translate/) for open reading frame (ORF), Compute pI/MW tool (https://web.expasy.org/compute_pi/) for molecular weight (MW) and isoelectric point (pI) and Simple Modular Architecture Research Tool GENOMES (*Letunic, Khedkar & Bork, 2021*) (http://smart.embl-heidelberg.de/smart/set_mode.cgi?GENOMIC=1) for functional protein domains. Homologous amino acid sequences in other crustaceans and arthropods (Tables 1 and 2) were obtained from NCBI GenBank database with literature and BLAST searches (https://blast.ncbi.nlm.nih.gov/). Selected species included in the analyses were those in evolutionary lineages leading to *P. monodon* and with putative orthologs of Pmtra

2, PmOvtra 2, Pmfru-1 and Pmfru-2 proteins, and their homologs/orthologs copies were either functionally characterized in other species (literature searches) or with highest sequence similarity to the corresponding homologs of interest in *P. monodon* (BLAST searches). Multiple sequence alignments of the homologous amino acid sequences of Tra 2 and Fru gene sets were separately performed with MAFFT version 7 web service (*Katoh, Rozewicki & Yamada, 2019*) with a default setting (https://mafft.cbrc.jp/), the best substitution model for each protein sequence set was obtained with ProtTest 3 (version 3.4.2 (*Darriba et al., 2011*)), and the phylogenetic trees were performed with RAxML (version 8.2.12; (*Stamatakis, 2014*)) with 1,000 bootstrapping. Tra 2 phylogenetic tree was constructed with a GAMMA model of rate heterogeneity and the WAG substitution model with empirical amino acid frequencies, but Fru phylogenetic tree was constructed with a GAMMA model of rate heterogeneity and the VT substitution model with empirical amino acid frequencies.

## Animal and tissue sampling

Male and female *P. monodon* broodstocks were obtained from Shrimp Genetic Improvement Center (SGIC), Suratthani, Thailand. All shrimps were tested for specific pathogen infections, *i.e.,* White spot syndrome virus (WSSV), Yellow head virus (YHV), Taura syndrome virus (TSV), Monodon baculovirus (MBV), *P. monodon* densovirus (PmDNV), Infectious hypodermal and hematopoietic necrosis virus (IHHNV), Decapod iridescent virus 1 (DIV1) and *Vibrio parahaemolyticus.* Shrimp were acclimatized in 1000 L artificial seawater (30–35 ppt) in fiberglass tanks with constant aeration. The water temperature was maintained at 28 ± 2 °C. Shrimp were fed with commercial pellets, three times a day. Shrimp were euthanized by ice water immersion before tissue collection. A variety of tissues, including eyestalk, nervous tissue, gill, hepatopancreas, heart, gastrointestinal tract, testis and ovary were collected from female and male shrimp. The tissue samples were immediately kept in Tri-Reagent® (MRC, USA) and stored at −80 °C until used. In addition, various developmental stages of larvae, including nauplii, protozoea, mysis, and postlarvae, were obtained from SGIC. Samples were collected, washed with Phosphate buffered saline (PBS), and preserved in 1 ml of Tri-Reagent® solution and stored at −80 °C until used. All animals experiment was approved by the Animal Ethics Committee, Faculty of Science, Mahidol University (MUSC63-019-527).

## RNA extraction and cDNA synthesis

Total RNA was extracted from the tissue samples using Tri-Reagent® according to the manufacturer's instruction. Briefly, the tissue samples were homogenized in Tri-Reagent® solution and incubated at room temperature for 5 min. Homogeneous mixture was subjected to chloroform extraction and the total RNA was precipitated with isopropanol. The RNA pellet was washed by 80% and 100% ethanol and finally RNA sample was suspended in DEPC treated sterile distilled water. Concentration of RNA was determined by Nanodrop spectrophotometer (ThermoScientific, USA). For cDNA synthesis, DNaseI was used to remove genomic DNA in the samples. One microgram of the total RNA was treated with 1 U DNaseI (Thermo Fisher Scientific, Denmark), and the reaction was performed at

Thaijongrak et al. (2022), *PeerJ*, DOI 10.7717/peerj.12980

**Table 1 Accession numbers of Tra 2 homologous amino acid sequences used for multiple sequence alignment and phylogenic tree analyses.**

| Species | Subphylum/Class | Order | Family | Accession number | Gene name | |
|---|---|---|---|---|---|---|
| | | | | | with functional study (Reference) | putatively annotated without functional study |
| *Penaeus monodon* | Crustacea/Malacostraca | Decapoda | Penaeidae | MT543028 | PmOvtra 2 (this study) | |
| *Penaeus monodon* | Crustacea/Malacostraca | Decapoda | Penaeidae | ACD13597 | Pmtra 2 (*Leelatanawit et al., 2009*) | |
| *Penaeus chinensis* | Crustacea/Malacostraca | Decapoda | Penaeidae | AFU60541.1 | Pctra 2b (*Li et al., 2012*) | |
| *Penaeus vannamei* | Crustacea/Malacostraca | Decapoda | Penaeidae | XP_027230515.1 | | Transformer-2 protein homolog alpha-like |
| *Penaeus vannamei* | Crustacea/Malacostraca | Decapoda | Penaeidae | XP_027226065.1 | | Transformer-2 protein homolog alpha-like isoform X2 |
| *Scylla paramamosain* | Crustacea/Malacostraca | Decapoda | Portunidae | QNS26380.1 | Sptra 2 (*Wang et al., 2020*) | |
| *Cherax quadricarinatus* | Crustacea/Malacostraca | Decapoda | Parastacidae | QIH97833.1 | Cqtra 2b (*Cai et al., 2020*) | |
| *Eriocheir sinensis* | Crustacea/Malacostraca | Decapoda | Varunidae | APJ36535.1 | Estra 2c (*Luo et al., 2017*) | |
| *Homarus americanus* | Crustacea/Malacostraca | Decapoda | Nephropidae | KAG7172586.1 | | Transformer-2 protein beta-like |
| *Macrobrachium nipponense* | Crustacea/Malacostraca | Decapoda | Palaemonidae | QBS13802.1 | Mntra 2a (*Wang et al., 2019*) | |
| *Macrobrachium rosenbergii* | Crustacea/Malacostraca | Decapoda | Palaemonidae | QBY91826.1 | Mrtra 2a (*McMillan, 2018*) | |
| *Drosophila melanogaster* | Insecta | Diptera | Drosophilidae | NP_476764.1 | Dmtra 2 (*Amrein, Gorman & Nöthiger, 1988*) | |

Thaijongrak et al. (2022), *PeerJ*, DOI 10.7717/peerj.12980

**Table 2  Accession numbers of Fru homologous amino acid sequences used for multiple sequence alignment and phylogenic tree analyses.**

| Species | Subphylum/Class | Order | Family | Accession number | Gene name | |
|---|---|---|---|---|---|---|
| | | | | | with functional study (Reference) | putatively annotated without functional study |
| *Penaeus monodon* | Crustacea/Malacostraca | Decapoda | Penaeidae | MT497519 | Pmfru-1 (this study) | |
| *Penaeus monodon* | Crustacea/Malacostraca | Decapoda | Penaeidae | MT503286 | Pmfru-2 (this study) | |
| *Penaeus vannamei* | Crustacea/Malacostraca | Decapoda | Penaeidae | XP_027234425.1 | | Longitudinals lacking protein, isoforms H/M/V-like isoform X1 |
| *Eriocheir sinensis* | Crustacea/Malacostraca | Decapoda | Varunidae | ART29432.1 | Esfru-2 (*Li et al., 2017*) | |
| *Eriocheir sinensis* | Crustacea/Malacostraca | Decapoda | Varunidae | ART29431.1 | Esfru-1 (*Li et al., 2017*) | |
| *Hyalella azteca* | Crustacea/Malacostraca | Amphipoda | Hyalellidae | XP_018020859.1 | | Sex determination protein fruitless-like isoform X1 |
| *Daphnia magna* | Crustacea/Branchiopoda | Cladocera | Daphniidae | XP_032789566.1 | | Longitudinals lacking protein, isoforms F/I/K/T-like |
| *Daphnia pulex* | Crustacea/Branchiopoda | Cladocera | Daphniidae | EFX71514.1 | | Hypothetical protein |
| *Nasonia vitripennis* | Insecta | Hymenoptera | Pteromalidae | NP_001157594.1 | Nvfru (*Bertossa, Van de Zande & Beukeboom, 2009*) | |
| *Aedes aegypti* | Insecta | Diptera | Culicidae | AGC11799.1 | Aafru (*Salvemini et al., 2013*) | |
| *Drosophila melanogaster* | Insecta | Diptera | Drosophilidae | NP_732344.1 | Dmfru (*Ryner et al., 1996*) | |
| *Drosophila melanogaster* | Insecta | Diptera | Drosophilidae | AAB96677.1 | Dmfru (*Ryner et al., 1996*) | |
| *Araneus ventricosus* | Chelicerata | Araneae | Araneidae | GBM78970.1 | | Protein bric-a-brac 2 |
| *Nymphon striatum* | Chelicerata | Pantopoda | Nymphonidae | KAG1658585.1 | | Protein bric-a-brac 2 |

**Table 3  Specific primers used in this study.**

| Name | Primers sequence (5′→ 3′) | Description | PCR product size |
|------|---------------------------|-------------|------------------|
| PmOvtra 2 F | AGTTCGTGAAAGGTCGAGGG | RT-PCR, qPCR | 449 bp |
| PmOvtra 2 R | CAGTACACTGCTCCTTGGCT | RT-PCR, qPCR | |
| Pmfru-1 F | GGAGGCAAGCATCAGTTTCG | RT-PCR, qPCR | 262 bp |
| Pmfru-1 R | AGATCGTTTGGGTCCCCTCA | RT-PCR, qPCR | |
| Pmfru-2 F | AGGCAGCTGACAACAATGCT | RT-PCR, qPCR | 345 bp |
| Pmfru-2 R | TTTTGGGTTATGAGGTGTGCCT | RT-PCR, qPCR | |
| 16s rRNA F | TGACCGTGCRAAGGTAGCATA | RT-PCR, qPCR | 152 bp |
| 16s rRNA R | TTTATAGGGTCTTATCGTCCC | RT-PCR, qPCR | |

37 °C for 1 h. The reaction was terminated by EDTA following the manufacturer's protocol. The extracted RNA was added to reverse transcription reaction which consisted of Random hexamer (Invitrogen, USA) and SuperScript® III Reverse Transcriptase (Invitrogen, USA). The reaction was performed according to the manufacturer's instructions. All cDNA samples were stored at −80 °C until used.

## Reverse transcription polymerase chain reaction (RT-PCR)

Shrimp tissues, including eyestalk, nervous tissue, gill, hepatopancreas, heart, gastrointestinal tract, testis and ovary were collected from six healthy adult shrimp. Pooled samples were prepared for RT-PCR analysis. Expression patterns of *PmOvtra 2*, *Pmfru-1* and *Pmfru-2* in various tissues of adult males and females were investigated by RT-PCR. The specific primers were designed and shown in Table 3, and *16s rRNA* was used as housekeeping gene. Reverse transcription PCR master-mix included cDNA template, 10×PCR buffer, 10 μM of forward primer and reverse primer, 10 mM dNTP, 2.5 U of Taq DNA polymerase (Invitrogen, USA) and DEPC treated sterile distilled water. The amplifications were carried out using temperature cycling conditions as follows: 1 cycle of 94 °C for 5 mins; 35 cycles of 94 °C for 30s, 55 °C for 30 s and 72 °C for 45 s; 1 cycle of 72 °C for 10 min. The PCR products were resolved by 1.5% agarose gel electrophoresis.

## Quantitative real-time PCR (qPCR)

Developmental stages of larvae, including nauplii, protozoea, mysis, and postlarvae were collected and approximately 30 individuals in the same stage were pooled as one sample. Ovaries and testes were collected from five adult females and five adult males, and each was run as individual sample. Three replicates were prepared for each sample. Quantitative real-time PCR was performed in 96-well plates in a final volume of 20 μl. Each reaction contained 2 μl of cDNA, 10 μl of KAPA SYBR®FAST qPCR Master-Mix (2X) (KAPA Biosystems), 0.4 μl of specific forward and reverse primers (10 μM/ μl) (List of specific primers are shown in the Table 3), and 7.2 μl of DEPC treated sterile distilled water. The reaction was carried out in the 7500 Real Time PCR system (Applied Biosystems) with the following qPCR conditions: initial denaturing at 95 °C for 3 min; 40 cycles of 95 °C for 3 s, 60 °C for 30 s, and 72 °C for 30 s and finally, 1 cycle of 95 °C for 15 s, 60 °C for 1 min, 95 °C for 30 s, 60 °C for 15 s. The production of specific products was confirmed by

performing a melting curve analysis of the samples. The reaction without cDNA was used as the negative control. The same qPCR profile was performed for *16s rRNA* amplification which served as a reference gene. To confirm that *16s rRNA* was qualified as a reference gene used, the stability of *16s rRNA, EF1-α,* and *β-actin*, was evaluated in the samples. The result suggested that *16s rRNA* would be one of the versatile reference genes for this assay (Supplement 3 and Supplement 4). Data were analyzed with 7500 software v.2.3 (Applied Biosystems). The baseline was set automatically by the software to maintain consistency. The comparative CT method ($2^{-\Delta\Delta CT}$ method) was performed for analyze the expression levels of *PmOvtra 2*, *Pmfru-1* and *Pmfru-2* (*Livak & Schmittgen, 2001*).

## Statistical analysis

Data represented the mean ± standard deviation (S.D.). Statistical analysis was performed using Graph-Pad Prism 5 software (*Motulsky, 2007*). Statistical significance between two groups (testis and ovary) were determined using unpaired $t$-test. Significant difference was considered at $P < 0.01$. Statistical significance between groups of larvae developmental stages was determined using Tukey's multiple comparison tests and one-way ANOVA. Significant difference was considered at $p < 0.05$.

# RESULTS

## Molecular identification of PmOvtra 2 and Pmfru

The full-length cDNA sequence of *P. monodon ovary associated transformer 2* (*PmOvtra 2*) and two isoforms of *P. monodon fruitless*-like gene (*Pmfru-1 and Pmfru-2*) were obtained from our transcriptome database. *PmOvtra 2* (GenBank Acc. No. MT543028) included 1,774 nt. with 110 bp 5′-untranslated region (UTR), a 744 bp open reading frame (ORF) and 920 bp 3′-UTR (Supplement 1). The putative amino acid sequences analyses of PmOvtra 2 showed that PmOvtra 2 was comprised of 247 amino acids with predicted molecular weight of 28.41 kDa and the theoretical isoelectric point was 11.34. PmOvtra 2 contained a predicted RRM domain and two RS regions that were highly conserved among Tra 2 proteins in several species. The predicted RRM domain of PmOvtra 2 was located at 101–180 aa (Fig. 1).

The full-length cDNA sequences of *Pmfru-1* (GenBank Acc. No. MT497519) and *Pmfru-2* (GenBank Acc. No. MT503286) were 1,306 and 1,858 bp, respectively. *Pmfru-1* consisted of a 101 bp 5′-UTR, a 1,182 bp ORF and a 23 bp 3′-UTR. The cDNA sequence of *Pmfru-2* consisted of a 92 bp 5′-UTR, a 1,437 bp ORF and a 329 bp 3′-UTR (Supplement 2). The putative amino acid sequences of Pmfru-1 comprised of 393 amino acids with predicted molecular weight of 43.93 kDa, whereas the Pmfru-2 comprised of 478 amino acids with predicted molecular weight of 52.41 kDa. The theoretical isoelectric points of Pmfru-1 and Pmfru-2 were 5.38 and 5.56, respectively. Translated amino acid sequences of both Pmfru-1 and Pmfru-2 contained BTB domain with zinc finger domain which were highly conserved in Fru proteins in several insect and crustacean species. The BTB domain and zinc finger domain of Pmfru-1 were located at 31-126 aa and 339-389 aa, respectively, while that of Pmfru-2 were at 31-126 aa and and 373-395 / 401-424 aa (Fig. 2).

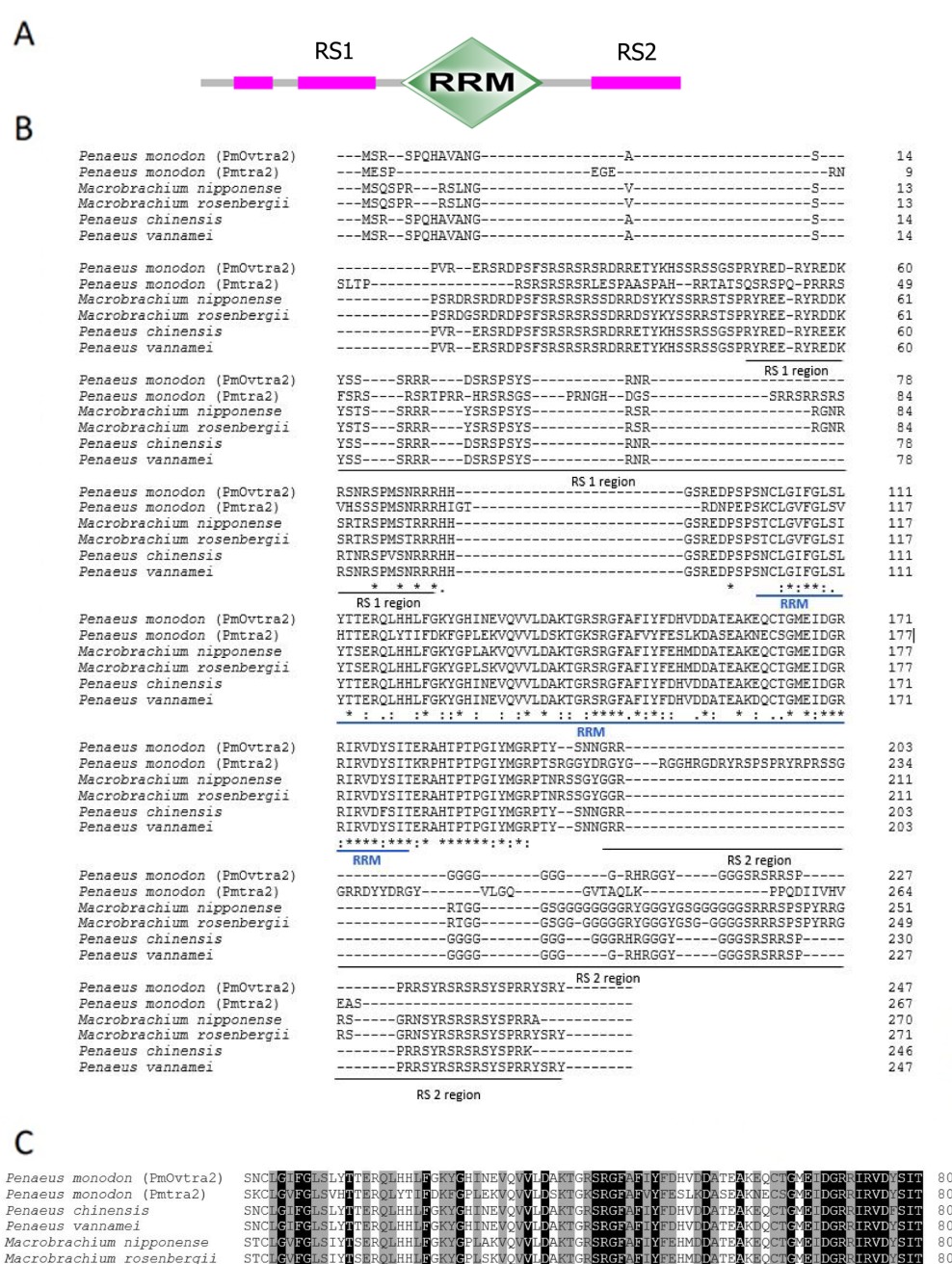

**Figure 1  Putative amino acid sequences with predicted domains of PmOvtra 2.** (A) Schematic presentation of amino acid sequences with predicted domains of PmOvtra 2 showing RRM domain and two RS regions which are conserved among Tra 2 homologs. (B) Deduced amino acid sequences with predicted domains of PmOvtra 2. The conserved sequences in the part of RRM domain were located at 103-176 aa in PmOvtra 2. (C) Multiple alignments of the RRM domain and two RS regions of PmOvtra 2 with Tra 2 homologs from other species showing highly conserved residues in all Tra 2 homologs. Conserved residues in all homologs were marked with black shadow. Partly conserved residues were marked with gray shadow.

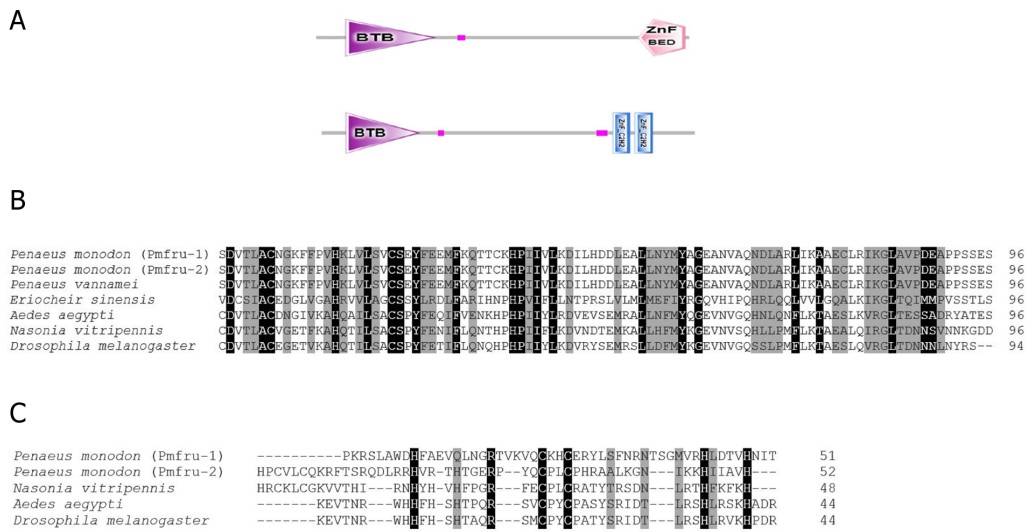

**Figure 2** **Putative amino acid sequences with predicted domains of Pmfru-1 and Pmfru-2.** (A) Schematic presentation of amino acid sequences with predicted domains of Pmfru-1 and Pmfru-2 showing BTB domain and zinc finger domain which are conserved among Fru homologs. The BTB domain and zinc finger domain of Pmfru-1 are located at 31-1-26 aa and 339–389 aa, respectively, while that of Pmfru-2 are at 31–126 aa and 373–395/401–424 aa. (B) and (C) Multiple alignments of BTB domain and zinc finger domain of Pmfru-1, Pmfru-2, and Fru homologs from other species. Conserved residues in all homologs were marked with black shadow. Partly conserved residues were marked with gray shadow.

## Multiple sequence alignments and phylogenic analysis of PmOvtra 2, Pmfru-1 and Pmfru-2

Due to evolutionary complexity of Tra 2 and Fruitless protein families, we focused on analyzing amino acid sequences of Pmtra 2, PmOvtra 2, Pmfru-1 and Pmfru-2 in *P. monodon* and of their homologs/orthologs (see Tables 1 and 2) in other species. Other putative paralogous proteins annotated in a draft genome sequence of *P. monodon* were not included because in this study we sought to infer evolutionary scenarios of Pmtra 2, PmOvtra 2, Pmfru-1 and Pmfru-2 but not of all paralogs in these two protein families (see Materials and Methods).

Tra 2 phylogenetic tree from maximum likelihood analyses of Tra 2 amino acid sequences (Fig. 3) revealed two distinct sister subclades of PmOvtra 2 and Pmtra 2 with their corresponding homologous sequences in the other crustaceans, and these two subclades containing PmOvtra 2 and Pmtra 2 formed a clade that was separated from *D. melanogaster* homolog (bootstrap re-sampling value 100%). The PmOvtra 2 and Pmtra 2 topology in the Tra 2 phylogenetic tree (Fig. 3) suggests that they were paralogous and duplicated in the last common ancestor of crustaceans. We named the two sister subclades containing PmOvtra 2 and Pmtra 2 paralogs as the PmOvtra 2 and Pmtra 2 clades, respectively (Fig. 3). Based on Tra 2 multiple amino acid sequence alignment analyses, PmOvtra 2 had higher percentage amino acid identities to the other previously reported Tra 2 orthologs in crustaceans within the PmOvtra 2 clade (98.68%, 89.47%, 86.18% and 86.18% to

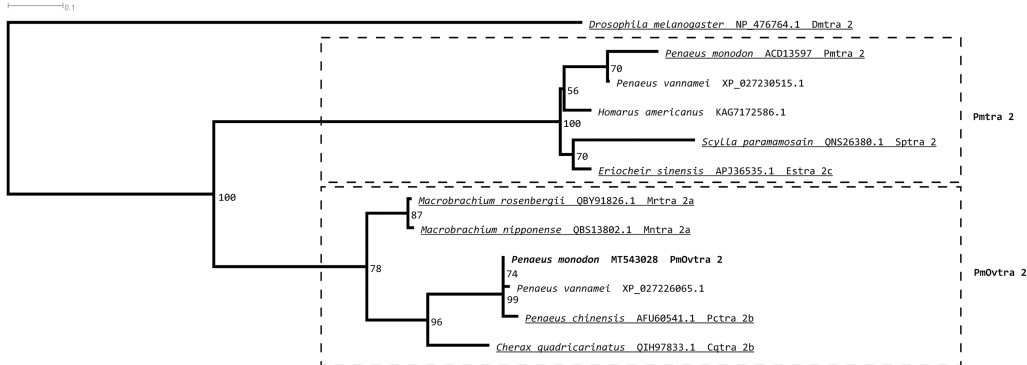

**Figure 3** **Phylogenetic placements of PmOvtra 2 and PmTra 2 on a maximum likelihood tree of Tra2 homologous sequences in arthropods and crustaceans.** Two separated sister clades containing PmOvtra 2 and PmTra 2 are clearly indicated by two dashed-line rectangles. Previously functional characterized sequences are underlined and with gene names. Bootstrap values (expressed as percentages of 1,000 replications) are near branching points, and *Drosophila melanogaster ortholog* was used as an outgroup.

*P. chinensis, C. quadricarinatus, M. rosenbergii* and *M. nipponense*, respectively) than to the previously characterized paralog Pmtra 2 (62.83%). Both PmOvtra 2 and Pmtra 2 had a high percentage amino acid identity (98.68% and 99.32%, respectively) to their putative ortholog in the *P. vannamei* draft genome. Similarly, for the conserved RRM domain (80 residues), PmOvtra 2 had a higher percentage identity to its orthologs in *P. chinensis* (98.75%), *P. vannamei* (98.75%) and *C. quadricarinatus* (92.50%) than to the paralog Pmtra 2 (70%). Note that similar percentage amino acid identities to the well-characterized insect *D. melanogaster* ortholog (264 residues) were observed for both PmOvtra 2 (51.97%) and Pmtra 2 (48.64%).

A maximum likelihood phylogenetic tree (Fig. 4) of Pmfru-1 and Pmfru-2 protein homologs (Table 2) placed both Pmfru-1 and Pmfru-2 with a copy in *P. vannamei* (bootstrap re-sampling value 100%) and in turn placed this clade of three proteins as a sister clade of another clade containing previously characterized Fru proteins in *E. sinensis* and insects, *e.g.*, Fru in *D. melanogaster* (bootstrap re-sampling value 85%). Such a placement of Pmfru-1 and Pmfru-2 in the phylogenetic tree suggested that the genomic locus (see below) of Pmfru-1 and Pmfru-2 was orthologous to that of a putative Fru copy in *P. vannamei* and had an evolutionary relationship with other previously characterized *fruitless* in *E. sinensis* and insects. Based on Fru multiple amino acid sequence alignment analyses, Pmfru-1 and Pmfru-2 had 99.31% amino acid identity to putative *P. vannamei* Fru copy, but 26.3% to *E. sinensis* Fru copy, the only characterized copy in crustacean so far. Interestingly, Pmfru-1 and Pmfru-2 had 38.1–43.6% amino acid identities to those well-characterized copies in insects. The BTB domain sequences of Pmfru-1 and Pmfru-2 were identical (100% identity due to the identical nucleotide sequences in the region; see Supplementary alignment 2), and they were also identical to that of *P. vannamei* (Fig. 2). They had 46.80%, 43.61% and 41.48% identity with that of *N. vitripennis, D. melanogaster* and *A. aegypti*, respectively, but they had 27.65% to that of *E. sinensis* Fru copy. Contrary to the BTB domain, the ZF domain sequences of Pmfru-1 and Pmfru-2 shared only 26.31%

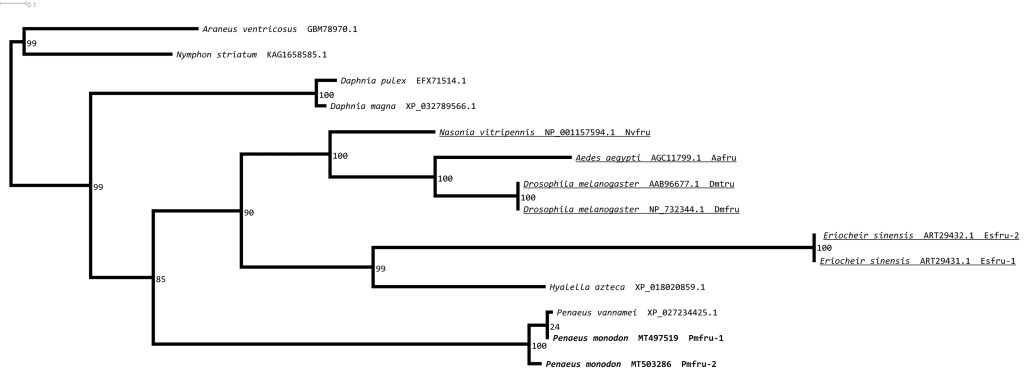

**Figure 4** **Phylogenetic placements of Pmfru-1 and Pmfru-2 on a maximum likelihood tree of Fru homologous sequences in arthropods and crustaceans.** Previously functional characterized sequences are underlined and with gene names. Bootstrap values (expressed as percentages of 1,000 replications) are near branching points, and Chelicerata orthologs (*Araneus ventricosus* and *Nymphon striatum*) were used as an outgroup.

identity. Pmfru-1 ZF domain sequences had a lower sequence identity to those of *D. melanogaster*, *A. aegypti* and *N. vitripennis* than did Pmfru-2 (28.94 *vs.* 31.57%, 28.94 *vs.* 31.57% and 23.68 *vs.* 31.57%, respectively). It should be noted that ZF domains were not predicted for the protein sequences of *P. vannamei* and *E. sinensis*.

## Gene organization and alternative splicing of Pmfru-1 and Pmfru-2

Since nucleotide sequences of *Pmfru-1* and *Pmfru-2* share identical sequences between 11-1109 nt of *Pmfru-1* and 2-1100 nt of *Pmfru-2* (see Supplementary alignment 2), we sought to determine whether *Pmfru-1* and *Pmfru-2* are results of alternative splicing events by searching the nucleotide sequences against a draft genomic sequence of *P. monodon*. Both *Pmfru-1* and *Pmfru-2* were aligned by BLASTN to only a single scaffold sequence (Accession number NC_051415.1 or PmonScaffold_30) and formed seven interval regions as putative seven exons of 87, 95, 594, 243, 90, 761 and 197 bp in length, respectively (Fig. 5). Five out of the seven exons showed the identical sequences between sequences of *Pmfru-1*, *Pmfru-2* and genomic scaffold sequences, whereas the other two (exons 3 and 6) each had one mismatched base-pair ($\geq$ 99.7% identity) to the genomic scaffold sequence. At these exon-intron junctions, splice dinucleotide sequences of GT donor sites and AG acceptor sites are observed. The result suggested that *Pmfru-1* and *Pmfru-2* were likely produced by constitutive splicing events of exons 1 to 5 and alternative splicing events of exon 6 to *Pmfru-2* mature mRNA and exon 7 to *Pmfru-1* mature mRNA (*i.e.,* mutually exclusive exons; Fig. 5). The start codons (ATG) of *Pmfru-1* and *Pmfru-2* were both located in exon 2, whereas the stop codons (TGA) of *Pmfru-1* and *Pmfru-2* were located in exon 7 and exon 6, respectively.

## Temporal expression profiles of PmOvtra 2, Pmfru-1 and Pmfru-2

The temporal expressions of *PmOvtra 2*, *Pmfru-1* and *Pmfru-2* were determined by qPCR. The specific primers were designed and shown in Table 3. The relative expression levels

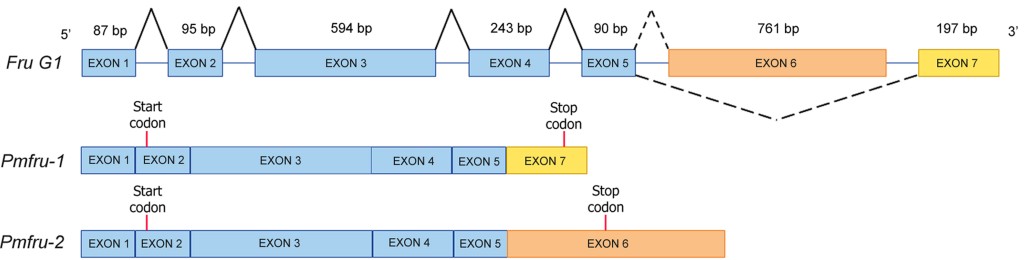

**Figure 5** Schematic diagram shows alternative splicing of *Pmfru-1* and *Pmfru-2* from the *Penaeus monodon* genomic region of seven exons. Exons 1 to 7 are of 87, 95, 594, 243, 90, 761 and 197 bp in length, respectively. Constitutive splicing and alternative splicing events are depicted by connecting solid and dashed lines, respectively. Mature *Pmfru-1* and *Pmfru-2* mRNAs share constitutive splicing exons 1–5, and alternative splicing events include mutually exclusive exon 6 for *Pmfru-2* and exon 7 for *Pmfru-1*. The start (ATG) and stop (TGA) codons are represented by red vertical bars on exon 2 and exon 7 & exon 6, respectively.

of *PmOvtra 2*, *Pmfru-1* and *Pmfru-2* were determined in different developmental stages of larvae and postlarvae, including nauplius, protozoea, mysis, and postlarvae stage 1, 5, 10, and 15 (PL1, PL5, PL10, and PL15). *PmOvtra 2*, *Pmfru-1* and *Pmfru-2* mRNA expression levels were normalized to the *16s rRNA* transcript level. The results showed that *PmOvtra 2* was detected in all the developmental stages of *P. monodon.* Notably, *PmOvtra 2* expression was relatively low in nauplius and protozoea stages, and then gradually increased from mysis to PL1. The *PmOvtra 2* expression level reached the highest level in PL1 but it abruptly decreased from PL5 to PL15 (Fig. 6A).

The *Pmfru-1* and *Pmfru-2* transcripts were detected in all larval stages of *P. monodon*. The *Pmfru-1* transcripts showed low expression levels at nauplii stage and slightly increased from protozoea to PL5. The *Pmfru-1* expression level reached the highest level at PL1 and gradually decreased from PL10 to PL15 (Fig. 6B). The *Pmfru-2* expression level maintained a low level from nauplius to mysis and then gradually increased at the postlarval stages. The *Pmfru-2* expression level reached high level in PL1 and then gradually decreased at PL5. The *Pmfru-2* transcripts also increased at PL10 and reached the highest levels at PL15 (Fig. 6C).

## Tissue distribution of *PmOvtra 2*, *Pmfru-1* and *Pmfru-2* in adult *P. monodon*

Expression of *PmOvtra 2*, *Pmfru-1*, and *Pmfru-2* transcripts were determined by RT-PCR in several tissues, including eyestalks, nervous tissues, gill, hepatopancreas, heart, gastrointestinal tract, and gonads. Expression of *16s rRNA*, showing the band at 152 bp, was utilized as internal control. The expression of *PmOvtra 2*, which exhibited the specific band at 449 bp, showed in nervous tissues, gill, heart, gastrointestinal tract, testis and ovary (Fig. 7). Apparently, the expression of *PmOvtra 2* was high in ovary compared with other tissues.

Expression of *Pmfru-1* exhibited the specific band at 262 bp. Reverse transcription PCR results showed that *Pmfru-1* was expressed in several tissue, *i.e.,* eyestalk, nervous tissues,

**A**

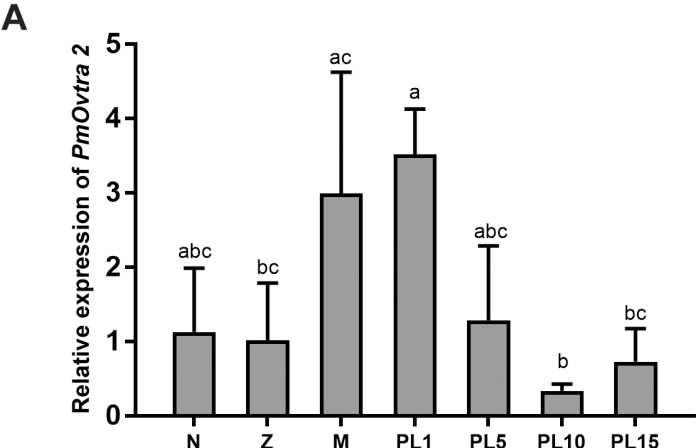

**B**

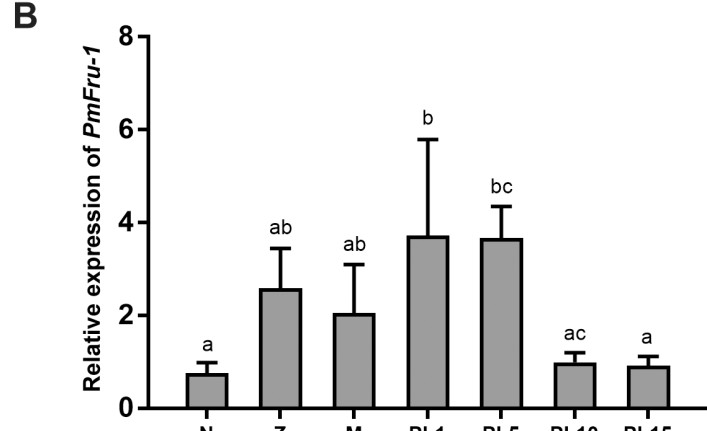

**C**

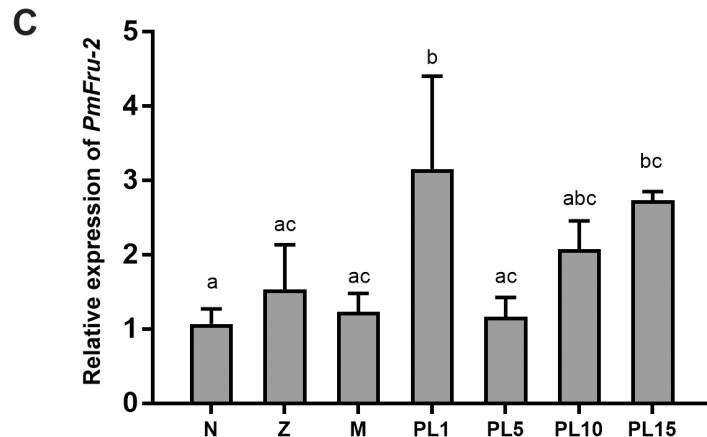

**Figure 6  Temporal expression of *PmOvtra 2, Pmfru-1,* and *Pmfru-2.* ** (A) Relative expression of *PmOv-tra 2.* (B) Relative expression of *Pmfru-1.* (C) Relative expression of *Pmfru-2.* The relative expression levels of these genes were analyzed by RT-qPCR in different developmental stages of *P. monodon* including nauplius (N), (continued on next page...)

**Figure 6 (…continued)**
protozoea (Z), mysis (M) and postlarvae (PL1, 5, 10, and 15). The *PmOvtra 2*, *Pmfru-1* and *Pmfru-2* mRNA levels were normalized to the *16s rRNA* mRNA level. Data represent the mean ± standard deviation (S.D.). Statistical significance was calculated by tukey's multiple comparison tests and one-way ANOVA. The significant difference was indicated by lettered bars ($p < 0.05$).

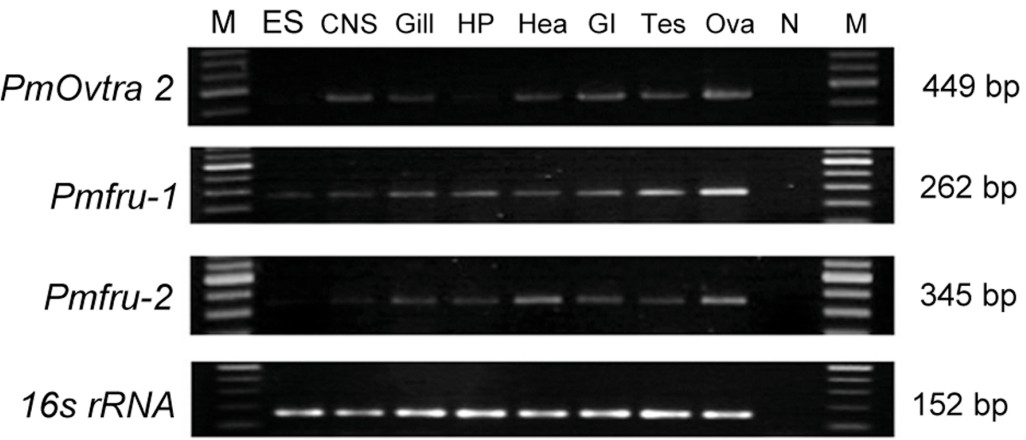

**Figure 7  Tissue distribution of *PmOvtra 2*, *Pmfru-1*, and *Pmfru-2* in *P. monodon*.** Expressions of *PmOvtra 2*, *Pmfru-1*, *Pmfru-2* were investigated in eyestalk (ES), central nervous system (CNS), gill, hepatopancreas (HP), heart (Hea), gastrointestinal tract (GI), testis (Tes), and ovary (Ova) by RT-PCR. M and N represented DNA markers and negative control, respectively. *PmOvtra 2*, *Pmfru-1* and *Pmfru-2* transcripts showed the specific band at 449 bp, 262 bp and 345 bp, respectively. The *16s rRNA*, showing the band at 152 bp, was utilized as internal control.

gill, hepatopancreas, heart, gastrointestinal tract, testis and ovary (Fig. 7). Notably, the band of *Pmfru-1* expression was intense in gonadal tissues, both testis and ovary. It was suggested that the gonad tissue might be a major organ for *Pmfru-1* mRNA expression. Regarding the expression of *Pmfru-2*, its expression exhibited the specific band at 345 bp. The result showed the expression of *Pmfru-2* in eyestalks, nervous tissues, gill, hepatopancreas, heart, gastrointestinal tract, and gonad, with high preference for ovary (Fig. 7).

Since *PmOvtra 2, Pmfru-1* and *Pmfru-2* were highly expressed in gonadal tissues as shown by RT-PCR, the expression patterns of these genes in testes and ovaries were further quantified by qPCR to see whether their expressions were different between sexes of *P. monodon*. The expression patterns of *PmOvtra 2*, *Pmfru-1*, and *Pmfru-2* in testes and ovaries were normalized to the *16s rRNA* transcript level. Relative expressions of *PmOvtra 2, Pmfru-1*, and *Pmfru-2* in the ovary were 26.37 ±10.64, 71.27 ±42.92, and 60.99 ±29.45, respectively. However, the relative expressions of these transcripts in the testis were much lower, *i.e.,* 1.20 ±0.41, 0.99 ±0.26 and 1.40 ±0.68, respectively. The result indicated significantly higher expression of *PmOvtra 2*, *Pmfru-1*, and *Pmfru-2* in the ovary compared with the testis ($p < 0.01$) (Fig. 8).

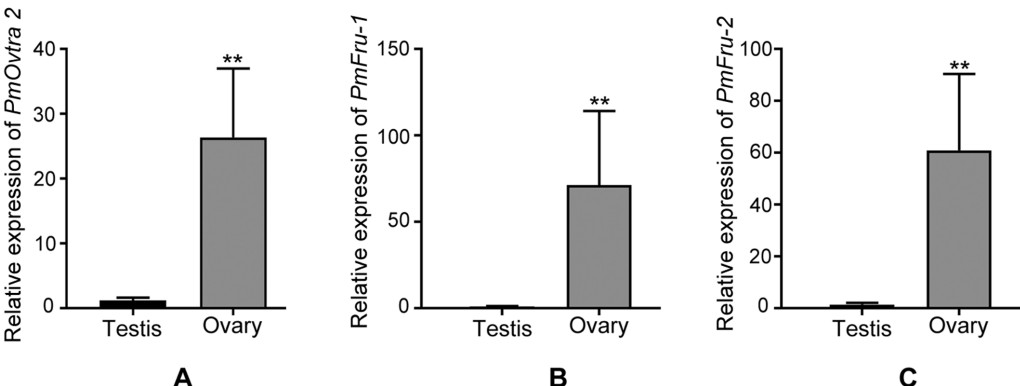

**Figure 8** **Relative expressions of *PmOvtra 2, Pmfru-1,*  and *Pmfru-2* in testes and ovaries revealed by qPCR.** (A) Relative expression of *PmOvtra 2.* (B) Relative expression of *Pmfru-1.* (C) Relative expression of *Pmfru-2.* The mRNA expression level of *PmOvtra 2, Pmfru-1* and *Pmfru-2* were normalized to the *16s rRNA* transcript level. The result showed greatly high expression level of *PmOvtra 2, Pmfru-1* and *Pmfru-2* in the ovary when compared with the testis. Data represented the mean ±standard deviation (S.D.). Statistical significance was calculated by unpaired t test. The significant difference was indicated by two asterisks (**) ($p < 0.01$).

## DISCUSSION

This study identified and characterized putative sex determining genes in *P. monodon*, including *PmOvtra 2, Pmfru-1*, and *Pmfru-2. Transformer 2* has been reported as a key gene for sex determination in some insects and crustaceans by regulating RNA splicing (*Cai et al., 2020*). Transformer 2 protein forms a heterodimer with Tra protein and acts as a splicing factor on downstream RNA that requires sex-specific splicing (*Nöthiger & Steinmann-Zwicky, 1985*). The direct targets of Tra 2 and Tra heterodimer include *doublesex* (*dsx*) and *fruitless* (*fru*) (*Haag & Doty, 2005*). In fact, *P. monodon transformer 2* (*Pmtra 2*; GenBank accession no. ACD13597) has previously been reported. In this study, a newly identified paralog of *Pmtra 2* in *P. monodon* was characterized and designated as *P. monodon ovarian associated tra 2* (*PmOvtra 2*) due to its pronounced expression in the ovary. Domain architecture of PmOvtra 2 shared highly conserved one RRM and two RS domains with Tra 2 proteins in insects and crustaceans (Fig. 1) (*Haag & Doty, 2005; Zhang et al., 2013a; Zhang et al., 2013b*). Our maximum likelihood phylogenetic tree of Tra 2 suggests the duplication event of PmOvtra 2 and the previously reported Pmtra 2 (*Leelatanawit et al., 2009*) most likely in the last common ancestor of crustaceans (Fig. 3), and each had its own orthologous sequences in almost all crustacean species with publicly available genomic sequences. Indeed, *PmOvtra 2* and *Pmtra 2* were found to be located on two different genomic loci/scaffolds in the draft *P. monodon* genome sequence, and they had different orthologs in *P. vannamei* (Table 1 and Fig. 3). Based on the amino acid sequence alignment, a higher similarity of either PmOvtra 2 or Pmtra 2 to their corresponding crustacean orthologs, for examples in *P. vannamei* (98.68% and 99.32%, respectively), than between PmOvtra 2 and Pmtra 2 (62.83%) also supports the duplication event of the two genes in the crustacean last common ancestor. Moreover, our maximum

likelihood phylogenetic tree of Tra 2 suggests that the previously reported Tra 2 in *P. chinensis, C. quadricarinatus, M. rosenbergii* and *M. nipponense* were orthologs of PmOvtra 2, and the previously reported Tra 2 in *E. sinensis* and *S. paramamosain* were orthologs of Pmtra 2 (Fig. 3). Note that gene names given to these previously characterized orthologs in crustacean species in each clades are varied, *e.g.*, the PmOvtra 2 clade contains Tra 2a and Tra 2b, and the Pmtra 2 clade contains Tra 2 and Tra 2c. Further studies are needed to characterize all paralogs in these crustacean genomes.

*Fruitless* gene is one of the key genes in sex determination in insects (*Bertossa, Van de Zande & Beukeboom, 2009*; *Gempe & Beye, 2011*). It belongs to the family of transcription factors BTB-Zn-finger, which represents a group of transcription factors (*Drapeau et al., 2003*; *Heinrichs, Ryner & Baker, 1998*; *Ryner et al., 1996*). In the present study, we discovered two *fruitless*-like homologs, namely *Pmfru-1* and *Pmfru-2*, in shrimp for the first time. The putative sequences of Pmfru-1 and Pmfru-2 exhibited highly conserved BTB domain and ZF domains. *Fruitless* have been identified in several insect and crustacean species, including many species in Drosophilidae, *Nasonia vitripennis*, *Gryllus bimaculatus*, and Chinese mitten crab *E. sinensis* (*Bertossa, Van de Zande & Beukeboom, 2009*; *Li et al., 2017*; *Ryner et al., 1996*; *Watanabe, 2019*; *Zollman et al., 1994*). We showed that *Pmfru-1* and *Pmfru-2* could be derived from the same genomic locus by mutually-exclusive-exons alternative splicing of the *Pmfru* pre-mRNA (Fig. 5). The two isoforms Pmfru-1 and Pmfru-2 were orthologous to a sequence in *P. vannamei* draft genome sequence with 99.31% amino acid identity. Based on our maximum likelihood phylogenetic tree analyses, the clade of Pmfru-1 and Pmfru-2 isoforms and their *P. vannamei* ortholog was clustered to the clade containing previously characterized Fru in *D. melanogaster*, *N. vitripennis*, and Chinese mitten crab *E. sinensis* (Fig. 4). Thus, the locus of Pmfru-1 and Pmfru-2 isoforms was conserved in penaeid shrimp, but its evolutionary relationship to the previously characterized *fruitless* in Chinese mitten crab *E. sinensis* needs to be further investigated since the placement of the two sister clades was supported by the bootstrap re-sampling value 85% (Fig. 4). Note that the current draft genome sequence of *P. monodon* contains other putatively annotated paralogs of *fruitless* (*e.g.*, on scaffolds NC_051406.1, NC_051418.1, NC_051427.1 and NC_051428.1). These genomic regions are different genomic regions from the region that was aligned with *Pmfru-1* and *Pmfru-2* isoforms on scaffold NC_051415.1 and that was putatively annotated for ORF of several longitudinals lacking protein isoforms. Evolutionary relationships and functional characterization among all paralogs of the BTB family proteins in *P. monodon* should be further investigated.

Functions of genes are commonly related to the pattern of gene expression. Therefore, the temporal and spatial expression of *PmOvtra 2*, *Pmfru-1*, and *Pmfru-2* transcripts were analyzed in this study. In our study, *16s rRNA* was used as an internal control gene in RT-PCR and qPCR assay (*Chotwiwatthanakun et al., 2016*; *Chotwiwatthanakun et al., 2018*). A study on reference genes for qPCR in *P. monodon* has suggested that *EF1-α* would be more versatile as a reference gene in reproductive organs (*Leelatanawit et al., 2012*). However, evaluation in our samples showed that *16s rRNA* and *β-actin* expressions in gonads were relatively stable (Supplement 3). Therefore, in this study, *16s rRNA* was used as an internal control gene in both gonads sample and various larvae and postlarvae development stages.

Expression of *PmOvtra 2* was detected in all the developmental stages of *P. monodon* starting from nauplius stage and it gradually increased from mysis to PL1 stages. Moreover, *Pmfru-1* transcripts showed low expression at nauplii stage and slightly increased from protozoea to PL5. The *Pmfru-2* expression level maintained at low level from nauplius to mysis and then gradually increased at the postlarval stages. Basically, genes involved in sex determination are expressed early during development before the appearance of genital organs. In *P. vannamei*, the genital organs, testis and ovary, are fully detected at 16 days postlarvae stage (*Garza-Torres, Campos-Ramos & Maeda-Martínez, 2009*). However, the differentiation of external gonad, including thelycum and gonopores, requires more time and they can be recognized around 50 days postlarvae stage (*Garza-Torres, Campos-Ramos & Maeda-Martínez, 2009*). In *P. japonicus*, the male reproductive organs, *i.e.,* testis, vas deferens and ejaculatory bulb, can be detected at 20 days postlarvae stage (*Nakamura, Matsuzaki & Yonekura, 1992*). In *P. chinensis*, the differentiation of gonad cannot be discriminated before PL76 (*Li et al., 2012*). In this study, *PmOvtra 2*, *Pmfru-1* and *Pmfru-2* transcripts could be detected at a very early stage of development which was earlier than the differentiation of gonad in *Penaeus* spp.

In crustaceans, the sex determination genes show sexually dimorphic expression pattern rather than sex-specifically alternative splicing (*Li et al., 2012*; *Li et al., 2018*; *Liu et al., 2015*). Dimorphic expression of many sex determination genes in gonads of crustaceans have also been reported, *e.g.*, *Pmtra 2* in *P. monodon* (*Leelatanawit et al., 2009*), *tra 2* in *P. chinensis*, *S. paramamosain*, *E. sinensis* (*Li et al., 2012*; *Luo et al., 2017*; *Wang et al., 2020*), *fru* in *E. sinensis* (*Li et al., 2017*), *dsx* in *D. magna* (*Kato et al., 2011*), *sxl* in *P. vannamei* and *M. nipponense* (*López-Cuadros et al., 2018*; *Zhang et al., 2013a*; *Zhang et al., 2013b*). This study also showed sexually dimorphic expression patterns of *PmOvtra 2, Pmfru 1*, and *Pmfru 2* genes in fully developed gonads. Notably, expression of *PmOvtra 2* in the ovary was greatly higher when compared with the testis of *P. monodon*, suggesting possible function of *PmOvtra 2* in the female sex determination pathway. The dominant expression of *PmOvtra 2* in ovary was similar to that of *Fctra-2c* in *P. chinensis* which was reported to be involved in female sex determination (*Li et al., 2012*). In contrast, in *M. nipponense*, *Mntra 2* was suggested to be involved in male sex determination as it displayed significantly higher expression level in testis than ovary (*Zhang et al., 2013a*; *Zhang et al., 2013b*). In *S. paramamosain*, *Sptra 2* was also highly expressed in several tissues of male crabs, suggesting its role in male sex determination system (*Wang et al., 2020*). In Chinese mitten crab *E. sinensis*, alternatively splicing of *tra 2*, *Estra-2a* and *Estra-2c*, has been reported and suggested to play role in male and female sex determination system, respectively (*Luo et al., 2017*). Role of *tra 2* in sex determination appears to be diverse in different species. In *P. monodon*, specific roles of *PmOvtra 2* in sex determination and ovary development need to be verified in further investigation.

In this research, we showed that expression of *Pmfru-1* and *Pmfru-2* was sexually dimorphic in the fully developed gonads. In adult male and female *P. monodon*, the expression levels of *Pmfru-1* and *Pmfru-2* in ovary was greatly higher than that in testis, suggesting the possible function in female sex determination pathway. Similarly, in *E. sinensis*, *Esfru1* expression has been shown particularly in ovary, suggesting its possible

function in female sex determination (*Li et al., 2017*). In contrast to *D. melanogaster*, Fru proteins play role in the determination of male sexual behavior (*Dauwalder, 2011*; *Gailey & Hall, 1989*; *Ryner et al., 1996*; *Salvemini, Polito & Saccone, 2010*; *Yamamoto, 2008*).

## CONCLUSION

In conclusion, this was the first time to report the existence of *fruitless*-like homologs in shrimp, *i.e.*, *Pmfru-1* and *Pmfru-2*, and also another paralog of *tra 2* in *P. monodon*, *PmOvtra 2*. All of them were greatly expressed in the ovary of *P. monodon*. These sexually dimorphic expression patterns in gonadal tissues suggested that *PmOvtra 2*, *Pmfru-1* and *Pmfru-2* may be involved in female-sex determination in this species. However, the detailed mechanism of sex determination in shrimp remains to be investigated.

### Funding

This research project was supported by the Mahidol University's Academic Development Scholarship (to Prawporn Thaijongrak), Mahidol University under the Fundamental Fund: Basic Research Fund: fiscal year 2022 (Grant no. BRF1-054/2565; to Rapeepun Vanichviriyakit), and the National Center for Genetic Engineering and Biotechnology (BIOTEC, NSTDA) of Thailand (to Anuphap Prachumwat). The funders had no role in study design, data collection and analysis, decision to publish, or preparation of the manuscript.

### Grant Disclosures

The following grant information was disclosed by the authors:
The Mahidol University's Academic Development Scholarship.
The Mahidol University under Fundamental Fund: Basic Research Fund: fiscal year 2022: BRF1-054/2565.
The National Center for Genetic Engineering and Biotechnology (BIOTEC, NSTDA) of Thailand.

### Competing Interests

The authors declare there are no competing interests.

### Author Contributions

- Prawporn Thaijongrak conceived and designed the experiments, performed the experiments, analyzed the data, prepared figures and/or tables, authored or reviewed drafts of the paper, and approved the final draft.
- Charoonroj Chotwiwatthanakun and Rapeepun Vanichviriyakit conceived and designed the experiments, analyzed the data, authored or reviewed drafts of the paper, and approved the final draft.
- Phaivit Laphyai and Thanapong Kruangkum performed the experiments, prepared figures and/or tables, and approved the final draft.

- Anuphap Prachumwat analyzed the data, prepared figures and/or tables, authored or reviewed drafts of the paper, and approved the final draft.
- Prasert Sobhon analyzed the data, authored or reviewed drafts of the paper, and approved the final draft.

### Ethics

The following information was supplied relating to ethical approvals (i.e., approving body and any reference numbers):

The Animal Ethics Committee, Faculty of Science, Mahidol University (MUSC63-019-527).

### DNA Deposition

The following information was supplied regarding the deposition of DNA sequences:

The nucleotide sequences of PmOvtra 2, Pmfru-1, and Pmfru-2 are available in the Supplementary Files and in GenBank: MT543028, MT497519, and MT503286, respectively.

### Data Availability

The cDNA sequences and predicted amino acid sequences of PmOvtra 2, Pmfru-1, and Pmfru-2, the raw data of Figures 6 and 8, the uncropped gels of Figure 7 are provided and the raw data of threshold cycles (CT) values of three reference genes used in qPCR analysis are available in the Supplemental Files.

### Supplemental Information

Supplemental information for this article can be found online at http://dx.doi.org/10.7717/peerj.12980#supplemental-information.

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
