# Peer review of "Molecular characterization and expression profiling of transformer 2 and fruitless-like homologs in the black tiger shrimp, Penaeus monodon"

_PeerJ, doi:10.7717/peerj.12980_

## Round 0.1 · original submission · Major Revisions

Reviewers have commented on your work, and I agree with them in their requests that you please kindly revise your work.

It is important for you to provide great detailed responses, not only in replying to the reviewers, but also in the text, yet succinctly. Looking forward to your revised manuscript. This is a very promising study.

Reviewer 1 ·

Basic reporting

The MS analyses in the sequences of two genes related to sex determination in insects, tra-2 and fru, as well as their phylogenetic position and their temporal and spatial expression patterns in P. monodon.

The authors suggest that both tra-2 and fru could be involved in sex determination pathway of Penaeus monodon, one of the most traded shrimp species. Previously, several pieces of evidence have been similarly reported that tra-2 and fru may play the same role in sex determination of some crustaceans as they do in insects. However, other studies have indicated the opposite. Therefore, given that sex determination/differentiation in crustaceans is still very puzzling, new findings related to this are very interesting, especially when they come from an aquaculture species.

The MS is well written and easy to read.
Figures and tables are well presented.
A little more background/context might help non-specialist readers to better understand the difficulties in studying sex-related genes in crustaceans.
Minor comments are included in the PDF version.

Experimental design

The MS describes the sequences of an ovarian tra-2 and two fru isoforms from P. monodon.

As for phylogenetic analyses, I would say there is much room for improvement. Maximum likelihood trees and/or Bayesian trees would be preferable to neighbor-joining trees built with MEGA. Detailed information is not provided either on codon partitioning schemes or on evolutionary models. Therefore, I encourage the authors to re-run the phylogenetic analyses using more sophisticated software to obtain more robust results. In addition, more both tra-2 and fru crustacean sequences could be added to the datasets to further clarify the phylogenetic relationships of P. monodon with other crustacean/insect species, if that is one of the goals. Sequences of these genes from taxa closer to P. monodon than to E. sinensis are currently available from NCBI and could be useful. Finally, I suggest that should specify which species will be used to root the trees appropriately.

As for the RT-PCR and RT-qPCR assays, they are well done. The only drawback is that they have only used one housekeeping gene (16S rRNA), while it is strongly recommended to use more than one reference gene, and even more so if it is a mitochondrial one. However, I assume that redoing these assays is not feasible at the moment.

Validity of the findings

Novel and interesting findings are provided, as new data will be useful to further unravel sex determination in crustaceans and in particular in P. monodon.

The expression of tra-2 and fru throughout the development of P. monodon is particularly interesting, as well as their differential expression between ovary and testis.

My main concern is the validity of the phylogenetic discussion inferred from the neighbor-joining trees. Indeed, the split between P. monodon Ovatra2 and an earlier P. monodon tra2 sequence is unsupported (bootstrap=48). Therefore, these clades are not feasible. I strongly recommend re-running these analyses under ML/BI to check this. Furthermore, based on the tra-2 tree, it is proposed that the new Ovatra2 sequence in P. monodon may be involved in sexual development and that it is a paralog of the former tra2 gene. Certainly, if this is the actual phylogenetic position of Ovatra2 and tra2, further discussion of the different genomic loci would be desirable in the Discussion section.

Additional comments

The sequences of Ovatra2 and fru isoforms were obtained from an unpublished transcriptome database. I wonder why the remaining components of the Sxl-Tra/Tra2-Fru/Dsx sex determination pathways were not also included in this work, especially in the case of the sex-lethal master gene. An explanation for this would be interesting

Finally, there is evidence that this pathway (including tra-2 and fru) could be involved in sex determination in several crustacean species as in P. monodon. Further references to such work would be desirable for the sake of clarity and also to support the findings of the present MS. At the same time, there is literature arguing that tra-2 and fru have not shown any evidence of participating in sex determination of other Decapoda (see review in Chandler et al. 2017). Therefore, I suggest authors to include more references for and against tra-2 and fru as sex regulators among different decapods to illustrate the crux of sex determination in Crustacea.

Annotated reviews are not available for download in order to protect the identity of reviewers who chose to remain anonymous.

Reviewer 2 ·

Basic reporting

no comment

Experimental design

no comment

Validity of the findings

no comment

Additional comments

The authors report the full cDNA of three transcripts which were named PmOvtra2, Pmfru-1 and Pmfru-2. These transcripts were obtained from a transcriptome analysis that the authors claims to have performed. In this work the authors also evaluated the expression patterns of these genes and also quantified their relative abundances. Although a previous tra2 gene had been reported in this species, the authors argue that PmOvtra2 is different than the previously published. Into this context, I believe the current work gives new information about the likely participation of these genes in the sex differentiation pathways in this shrimp. However, I find some issues that need to be further addressed by the authors in order to improve the manuscript.
Introduction:
The authors point out works performed in the M. nipponense prawn but they are leaving out works done in one of the most worldwide cultivated shrimp, Penaeus vannamei. I suggest to the authors to check the following works which could give a broader perspective of sex differentiations genes that have been reported in shrimps: Lopez-Cuadros et al., 2018 (Isolation of the sex-determining gene Sex-lethal (Sxl) in Penaeus (Litopenaeus) vannamei (Boone, 1931) and characterization of its embryogenic, gametogenic, and tissue-specific expression). Galindo-Torres et al., 2019 (A natural antisense transcript of the fem-1 gene was found expressed in female gonads during the characterization, expression profile, and cellular localization of the fem-1 gene in Pacific white shrimp Penaeus vannamei). Even the authors did not mention a work performed by Robinson et al. (2014) in which the authors found that in P. monodon, the ortholog to the C. elegans fem-1 sex differentiation gene, mapped to the position where a sex-determining locus was predicted by a QTL analysis. I also think the authors need to go deeper in the chromosomal setting that shrimps seem to have (see: Li et al., 2003; Robinson et al., 2014; Staelens et al., 2008; Zhang et al., 2007; Yu et al., 2017; Perez-Enriquez, et al., 2020; Wang et al., 2021).
Materials and methods:
Although it is possible to see in the supplementary files how many biological replicas were used, the authors should mention that in the methods section. The authors did not mention how many organisms were used in each step. How many biological replicates were set for qPCR? Did the authors perform pools at the end point PCR?. Did the authors use a single organism to run end point PCR? All this part lacks of clarity.
I found the qPCR section with some issues. It is important to take into account that reference gene expression is known to vary considerably depending on the biological scenario (i.e. different tissues, maturity stage, etc.). It’s for that reason that a reference gene-stability validation is always recommended for each experimental design. It is a common practice to normalize using pre-selected reference genes, but it does not mean it is the best way. In this case, the authors are assuming that the expression of 16s rRNA is stable among all the tissues evaluated (N,Z,M,PL1,PL5,PL10,PL15 and among ovary and testis). I would recommend select at least five reference genes and evaluate their stability through GeNorm, NormFinder, bestkeeper, etc. The authors claimed they have a transcriptome of this species where they could search for potential reference genes to be validated. At the end, the authors should identify what of those five genes were the most stable in the differents scenarios they are evaluating. Moreover, they should calculate efficiency in an individual way for all of the genes, because of in practice is well known that to assume 100% efficiency is not true at all (see Vandesompele et al., 2002). On the other hand, if the authors know a reference where it has been demonstrated that the expression of r16s is found stable in all of the tissues they evaluated they should point it out.
Result:
In the recently reported P. vannamei genome (Zhang et al.,2019), there are three proteins annotated as tra2 and tra-2C. I believe these proteins should be included into the analyses performed by the authors. In fact, when I perform a blastx of the PmOvtra, it hits against P. vannamei tra.
Discussion.
Due to the pathway of sex differentiation in shrimps seem to be a mixed of genes found in D. melanogaster and C. elegans pathways of sex differentiation, I consider the authors should enlarge the discussion content. For example, including works performed in P. vannamei or even in the same P. monodon (See references pointed out in the introduction).
Other minor points:
There isn’t an accession number for any of the Pm Fruitless isoform reported, why?
Line 303: The authors use “tra” to refer a protein. Generally, a name of protein is written with the first letter capitalized or all the letters capitalized like Tra or TRA protein. Is this correct?
Line 306: The authors mention the existence of a previous work in which a tra2 gene is reported. Why didn't the authors mention this work in the introduction section?
Line 315: What does mean PM?
General comments
Line 142-143: Add the Nanodrop company name.
Terminology needs further attention. I would suggest to the authors to change the terminology used to refer to real time PCR and end point PCR. See Bustin et al., 2009 (The MIQE Guidelines:Minimum Information for Publication of Quantitative Real-Time PCR Experiments).
Why do standard deviation bars are big in figure 8? It seems that few biological replicates were used or why do the dispersions are varying considerably?. When I see the supplementary files, the Cts of these genes are not varying so much, so, why do dispersions are big?

---

## Round 0.2 · Minor Revisions

Please kindly address all comments raised by the reviewer. Thank you

Reviewer 2 ·

Basic reporting

no comment

Experimental design

no comment

Validity of the findings

no comment

Additional comments

qPCR issues.
I still find some issues that need to be addressed by the authors. The authors performed a gene stability analyses using the standard deviation as an indicator. At the end they ended up using the 16s gene as the most stable gene even though actin gene showed the lowest sd value in both, ovary-testis, and during larvae development (supplementary file 3 and 4). Why did the authors use 16s gene when it showed the higher sd value in both valuations?. Moreover, the authors should add in the materials and method section if they performed a melting curve analysis to prove one product for each amplicon was got. Finally, the authors should add the description of the analysis they performed to conclude 16s was the most stable gene comparing against ef1 and actin in the qPCR method section.

Result and discussion section.
In the genome of P. monodon there are 8 proteins annotated as “sex determination protein fruitless-like”, from which 4 (XP_03779110{3..6}.1) are identical and three more show different protein sequence (XP_037772638.1, XP_037799023.1, XP_037799203.1 and XP_037799203.1). These proteins are found in a different scaffold than the fruit isoforms reported by the authors. In fact the sequence reported by the authors is found annotated as “longitudinals lacking protein”. I believe the authors should add these sequences to the phylogeny analysis or to mention why they weren’t included. The same for the Tra sequence, there are some Tra proteins annotated and some show slight differences. I believe the authors should discuss about this.


Minor points.
Line 38: Change “spiced” by spliced.
Line 57: split “Incrustace” to in crustace.
Line 76-77: Check if tra-tra2 interaction does not occur in males
Line 183: The authors say they used 5 adult females and males, but in supplementary data 3, only 4 biological samples appear.
Line 215: change “tra proteins” for Tra proteins.
Line 274: add the name of the scaffold.
Figure 3: I suggest to the authors to improve the quality of the figure and add the genes names for clarity. For example, there are two P. vannamei proteins but without gene name (are these tra-2C?), it is hard to follow the idea. The same for figure 4.
Line 323: The author should homologue terms, here they refer to qpcr, and in other parts of the document as RT-qPCR.
In table 3, the author should add the size of the PCR product as it’s simplest to see them in a table.

---

## Round 0.3 · Minor Revisions

Please, authors, the reviewers have considered your work, and require you to do further work on it. Kindly address all comments raised as diligently as you can, in the best detail. Looking forward to your revised manuscript.

Thank you very much.

Reviewer 1 ·

Basic reporting

The manuscript has improved after revision. However, I found it difficult to check whether the authors followed my earlier recommendations, as the rebuttal letter available to me includes the responses to the other reviewer and not to my comments. For next time, I recommend a single letter with the responses to all reviewers in order to avoid these problems.

My main concerns remain the methodology, interpretation, and discussion of the phylogenetic analyses.

Experimental design

The authors have replaced the neighbor-joining trees of the previous version with ML trees built in RaxML.

They used MEGA to determine the best substitution model. Nevertheless, the number of models included in MEGA is very limited, so I recommend other software, such us jModelTest. In addition, although tra-2 and fru are both coding genes, whether their three partitions follow the same evolutionary model or not is not explored with specific software (e.g. PartitionFinder).

Human sequences are used as outgroups, which I consider inappropriate. Outgroups should not be too far from the study group (crustaceans). Therefore, the non-crustacean sequences could serve perfectly well as outgroups. I would eliminate human sequences as they could cause biases in the analyses.

Validity of the findings

Correction of these aspects of the phylogenetic analysis could lead to more robust results. The separation between the ovatra-2 and tra-2 clades remains unsupported (bootstrap = 74 < 90). I believe that improving the methodology of phylogenetic analyses probably could provide more support for these clades, which is vital for discussing the results.

Another issue that concerns me is that throughout section 3.2 of the Results and in the Discussion, the authors refer bootstrap as “percentage identity” and “similarity”, when the bootstrap values in a tree are not that. Bootstrap is a statistical measure of the support that the nodes/clades have in a phylogeny. I recommend re-writing these sections to avoid referring to bootstrap support of phylogenetic trees in this way because it is a serious error.

Additional comments

Here are some minor issues:
- References to the software used in the Materials and Methods section are still missing.
- Carefully review the use of italics when referring to gene/protein sequences because I notice some inconsistency.

Reviewer 2 ·

Basic reporting

no comment

Experimental design

no comment

Validity of the findings

no comment

Additional comments

qPCR.
The authors claim that 16s is the most stable gene, but for example, in the supplement file 4, the lowest standard deviation was for actin. In the case of ovary and testis, the author should calculate this value using all the samples (ovary and testis) because they are comparing genes between these two tissues, so the reference gene should be stable in these two. Again if I calculate the sd in this case, the lowest value is assigned to the actin gene, which gives 0.97.


Minor points
line 74: Change sxl protein for Sxl protein.
Line 188: The author say they pooled around 30 individuals, is this correct? Or did they mean to say 3?

---

## Round 0.4 · Minor Revisions

Thank you for your patience as it has been challenging to secure reviewers. Please, reviewers, have considered your work favorably, however, one still raised concerns
.
Please, the editor encourages authors to carefully provide detailed explanations in the text, as well as reply particularly on:

- Why was 16s rRNA reference gene considered the most stable gene?

- Did the 16s rRNA gene show variations in expression and if yes, why did the authors select this gene to be tested for gene stability?

- You have claimed this gene might have a bias because of its function in male spermiogenesis, the SD values calculated show the gene appears not responding with a bias in gene expression. Why?

- Did you actually conduct the stability procedure as protocol? Was the goal actually to identify the most stable gene?

- Kindly strengthen the discussion regarding qPCR as a strong component of the work, in the context of the above. This is necessary

Please, provide a detailed response to all these. Thank you

Reviewer 1 ·

Basic reporting

No comment

Experimental design

No comment

Validity of the findings

No comment

Additional comments

Thank you for amending the MS following the comments.

Reviewer 2 ·

Basic reporting

no comment

Experimental design

no comment

Validity of the findings

no comment

Additional comments

The author claims that they desired to use 16s rRNA as a reference gene even when it wasn’t the most stable gene. The authors argue that this gene might have variations in expression since some particular function during spermiogenesis among others has been reported for this gene. First of all, if the author knew this why did the authors select this gene to be tested for gene stability?. Although the authors claim this gene might have a bias because of its function in male spermiogenesis, the SD values calculated by the authors are indicating that the gene isn’t responding with a bias in gene expression. If this claim were true, the SD value would be higher.
I didn’t find any reasonable reason why the author is using 16s rRNA. I get the impression the authors did the stability procedure just as protocol but not when the goal of identifying the most stable gene. Given the qPCR is a strong component of the manuscript, and that the qPCR lacks scientific rigor, I would reject the manuscript.

Reviewer 3 ·

Basic reporting

No comment

Experimental design

No comment

Validity of the findings

No comment

Additional comments

Overall, the authors have improved the manuscript following previous review comments. The current form can be accepted.

Regards

---

## Round 0.5 · accepted · Accept

I am very satisfied with the explanations the authors have provided, and am convinced this revised manuscript is acceptable for publication. Thank you authors for finding PeerJ as your journal of choice, and taking the time to undertake the rigors of the peer review process, which has elevated the quality of this scholarly piece of work. Looking forward to your future contributions. Congratulations